# Hope and Challenges: Immunotherapy in *EGFR*-Mutant NSCLC Patients

**DOI:** 10.3390/biomedicines11112916

**Published:** 2023-10-28

**Authors:** Dan Yan

**Affiliations:** 1Aflac Cancer and Blood Disorders Center of Children’s Healthcare of Atlanta, Atlanta, GA 30322, USA; dyan2@emory.edu; 2Department of Pediatrics, Emory University, Atlanta, GA 30322, USA

**Keywords:** non-small cell lung cancer, *EGFR* mutation, immunotherapy

## Abstract

EGFR tyrosine kinase inhibitors (TKIs) are the preferred initial treatment for non-small cell lung cancer (NSCLC) patients harboring sensitive *EGFR* mutations. Sadly, remission is transient, and no approved effective treatment options are available for EGFR-TKI-advanced *EGFR*-mutant NSCLCs. Although immunotherapy with immune checkpoint inhibitors (ICIs) induces sustained cancer remission in a subset of NSCLCs, ICI therapy exhibits limited activity in most *EGFR*-mutant NSCLCs. Mechanistically, the strong oncogenic EGFR signaling in *EGFR*-mutant NSCLCs contributes to a non-inflamed tumor immune microenvironment (TIME), characterized by a limited number of CD8^+^ T cell infiltration, a high number of regulatory CD4^+^ T cells, and an increased number of inactivated infiltrated T cells. Additionally, *EGFR*-mutant NSCLC patients are generally non-smokers with low levels of PD-L1 expression and tumor mutation burden. Promisingly, a small population of *EGFR*-mutant NSCLCs still durably respond to ICI therapy. The hope of ICI therapy from pre-clinical studies and clinical trials is reviewed in *EGFR*-mutant NSCLCs. The challenges of application ICI therapy in *EGFR*-mutant NSCLCs are also reviewed.

## 1. Introduction

Lung cancer is the leading cause of cancer-related death worldwide [1]. Non-small cell lung cancer (NSCLC) accounts for ~85% of all lung cancers. Up to 50% of NSCLCs in Eastern Asia and 8–16% of NSCLCs in the West had epidermal growth factor receptor (*EGFR*) mutations [2,3,4]. *EGFR* L858R missense mutation and in-frame exon 19 deletions (19del) were the most common mutations found in *EGFR*-mutant NSCLCs, accounting for ~85% of all *EGFR*-activating mutations [3,4,5,6]. NSCLCs with *EGFR* L858R or 19del were sensitive to initial EGFR tyrosine kinase inhibitor (TKI) therapy [7,8]. Relative to traditional chemotherapy, EGFR-TKI therapy significantly extended progression-free survival (PFS), overall survival (OS), and improved quality of life due to relatively low toxicity for NSCLC patients harboring sensitive *EGFR* mutations [9,10,11,12,13,14,15,16,17,18]. EGFR-TKI therapy was the first-line treatment for NSCLC patients with sensitive *EGFR* mutations in the National Comprehensive Cancer Network (NCCN) guidelines [19]. Unfortunately, the remission was transient for only nine months to around two years [9,10,11,12,13,14,15,16,20]. With the universal application of EGFR TKIs to treat the *EGFR*-mutant NSCLCs, EGFR-TKI resistance represents an unmet clinical problem. So far, no approved effective treatment options are available for EGFR-TKI-advanced *EGFR*-mutant NSCLCs, and these patients need chemotherapy-consisting therapy. However, the clinical benefit of chemotherapy-consisting therapy is modest, with a high potential for toxicity.

Other than targeted therapies, immunotherapy with immune checkpoint inhibitors (ICIs), represented by anti-programmed cell death-(ligand) 1 [anti-PD-(L)1] therapy, had demonstrated improved survival rates, long-term survival benefits, and a favorable safety profile over chemotherapy in a subset of advanced NSCLCs [21,22,23,24,25,26,27]. So far, there are six ICIs (ipilimumab, nivolumab, pembrolizumab, atezolizumab, avelumab, and durvalumab) approved by FDA as first- or second-line treatments for advanced NSCLCs but not suggested for patients with sensitive *EGFR* mutations [28]. Relative to NSCLCs with the *EGFR* wild-type, ICI therapy failed to improve survival benefits in most *EGFR*-mutant NSCLCs [29,30,31,32,33,34]. Clinical benefit from ICI therapy was associated with inflamed tumors with infiltrated immune cells, high tumor mutation burden (TMBs), and increased expression of PD-L1 on tumor cells [35]. However, the oncogenic EGFR signaling in lung cancer with *EGFR* mutations contributed to a non-inflamed tumor immune microenvironment (TIME) characterized by a limited infiltration of CD8^+^ T cells [36,37,38], a high number of infiltrations of regulatory CD4^+^ T cells [38], and an increased number of inactivated infiltrated T cells [39]. Additionally, *EGFR*-mutant NSCLCs were generally non-smokers [34] with low PD-L1 expression [36,37,39] and with a low tumor mutation burden [36,37]. The mechanisms mediating durable effects in a small proportion of *EGFR*-mutant NSCLCs were worth additional investigation [40].

## 2. Hope

### 2.1. The Efficacy of Osimertinib for EGFR-Mutant NSCLCs Is Independent of PD-L1 Levels

Osimertinib treatment decreased PD-L1 expression on *EGFR*-mutant NSCLC cell lines by reducing PD-L1 at the mRNA level and promoting PD-L1 degradation with proteasomes [41]. Interestingly, the median PFS (mPFS) of osimertinib was similar in PD-L1-negative patients (TC < 1%) (18.9 months) and in PD-L1-positive patients (TC ≥ 1%) (18.4 months) [42]. The data suggested that the osimertinib efficacy was independent of PD-L1 status in *EGFR*-mutant NSCLCs.

### 2.2. Subtypes of EGFR Alterations May Be Sensitive to ICI Therapy

ICI efficacy varies in the subtypes of *EGFR* alterations in *EGFR*-mutant NSCLCs. *EGFR* wild-type and L858R NSCLCs had a similar outcome, while *EGFR* 19del NSCLCs had a worse outcome when treated with ICI therapy [43]. Other groups also supported the findings that ICI therapy alone or in combination exhibited improved survival outcomes in the *EGFR* L858R group relative to the *EGFR* 19del group [44,45,46,47]. One of the possible explanations was that *EGFR* L858R-mutant tumors were more likely to have a significantly increased CD8^+^ T cell infiltration than *EGFR* 19del tumors [39]. Additionally, *EGFR* T790M-negative patients were more likely to benefit from ICI therapy than *EGFR* T790M-positive patients [48,49], as the *EGFR* T790M-negative tumors had higher PD-L1 expression than the *EGFR* T790M-positive tumors [48,50]. Other than the common mutations, 10–20% of *EGFR*-mutant NSCLCs harbor non-*EGFR* L858R or 19del mutations, known as uncommon mutations, such as S768I and G719X [5,37,51,52]. Relative to common *EGFR* mutations, uncommon *EGFR*-mutant tumors were likely to express high PD-L1 (≥50%), high TMB, and a high infiltration of CD8^+^ T cells [37,49,52,53]. Of the uncommon *EGFR*-mutant NSCLCs with dual high PD-L1 expression and abundant CD8^+^ T cell infiltration, 36.7% showed a favorable response to ICI therapy [49,53].

### 2.3. Heavily Pre-Treated EGFR-Mutant NSCLCs Are Likely to Respond to ICI Therapy Relative to Treatment-Naïve EGFR-Mutant NSCLCs

EGFR-TKI treatment results in dynamic changes in host immunity. EGFR-TKI-advanced *EGFR*-mutant NSCLC patients trended to have concurrent high levels of PD-L1 expression and a high number of CD8^+^ T cell infiltration [34]. The proportion of *EGFR*-mutant NSCLCs with high PD-L1 (≥50%) increased from 14% to 28% after EGFR-TKI treatment [54]. Heavily pre-treated EGFR-TKI-advanced *EGFR*-mutant NSCLCs with high expression of PD-L1 (≥25% of tumor cells) had a better median OS (mOS) than that with low expression of PD-L1 (<25% of tumor cells) (13.3 vs. 9.9 months) after ICI therapy in a Phase II ATLANTIC study [29,55]. Impressively, two of five EGFR-TKI-advanced *EGFR*-mutant NSCLCs with increased PD-L1 after an EGFR blockade showed a durable response to subsequent ICI therapy [54]. Overall, EGFR-TKI-advanced *EGFR*-mutant NSCLCs may be the groups responding to ICI therapy. However, ICI therapy immediately followed by EGFR TKI therapy may cause immune-related adverse effects (irAEs), as discussed below.

### 2.4. IrAEs Observed during ICI Therapy Are Associated with Efficacy

A growing body of literature supports the theory that the occurrence of irAEs during ICI therapy is associated with improved treatment efficacy. A retrospective analysis proved that the 270 metastatic NSCLC patients who experienced irAEs after ICI therapy had improved PFS, overall response rate (ORR), disease control rate (DCR), and OS relative to those who did not experience irAEs (PFS: 5.2 vs. 1.97 months; ORR: 22.9% vs. 5.7%; DCR: 76% vs. 58%; OS: not reached vs. 8.21 months) [56]. Similarly, secondary analysis of the Phase I CA209-003 clinical trial conducted at 13 US medical centers led to the finding that the mOS was significantly longer among patients with irAEs of any grade (19.8 months; 95% CI, 13.8–26.9 months) or grade 3 or more (20.3 months; 95% CI, 12.5–44.9 months) compared with those without treatment-related AEs (5.8 months; 95% CI, 4.6–7.8 months) (*p* < 0.001 for both comparisons based on hazard ratios) [21]. Another retrospective study found that around 43.6% of 195 advanced NSCLCs treated with ICI therapy experienced irAEs, and those NSCLCs with irAEs had significantly longer mPFS (5.7 s. 2.0 months), PFS (8.5 vs. 4.6 months), mOS (17.8 vs. 4.0 months), and OS (26.8 vs. 11.9 months) than those who did not experience irAEs [57]. In a prospective study with 43 advanced NSCLCs, earlier irAEs were associated with better objective response and disease control rates than those without irAEs after ICI therapy (37% vs. 17% and 74% vs. 29%, respectively) [58]. Another prospective study with 76 advanced NSCLCs also found that NSCLCs who experienced irAEs within two weeks of beginning ICI therapy had significantly longer mPFS than those who did not (5.0 vs. 2.0 months) [59]. However, it is challenging to predict irAEs, and severe irAEs may be life-threatening. A further understanding of irAEs during the application of immunotherapy is necessary.

### 2.5. Positive Clinical Studies Support the Use of ICI Therapy in EGFR-Mutant NSCLCs

Even though the ORR for *KRAS*-mutant NSCLCs to ICI therapy was around 20%, 7% of *EGFR*-mutant NSCLCs responded to single-agent ICI therapy [40]. A female NSCLC patient harboring an *EGFR* L858R mutation treated with ICI therapy resulted in a prolonged PFS (~23 months) [60]. Anti-PD-1/PD-L1 antibody-based ICI therapy, combined with other checkpoint inhibitors or platinum-based chemotherapy, has been widely applied in advanced lung cancers to achieve further improved outcomes. Yang et al. observed that chemo-immunotherapy was better than immunotherapy alone for EGFR-TKI-advanced *EGFR*-mutant NSCLCs (mPFS: 3.42 vs. 1.58 months, *p* = 0.027) [61]. A retrospective study of 122 EGFR-TKI-advanced *EGFR*-mutant NSCLC patients, especially with *EGFR* L858R mutation, revealed better mPFS (5.0 vs. 2.2 months) and mOS (14.4 vs. 7 months) in ICI-based combination therapy than that in ICI alone [45]. Furthermore, front-line ICI therapy reached better survival benefits than later-line ICI therapy [45]. In another retrospective study, chemo-immunotherapy also achieved better outcomes than ICI therapy alone for mPFS (4.3 vs. 1.5 months), mOS (14.92 vs. 7.41 months), and ORR (23.1% vs. 3.1%) in the EGFR-TKI-advanced *EGFR*-mutant NSCLCs, respectively [62]. Patients with EGFR L858R consistently exhibited a trend of longer mPFS (7.6 vs. 5.4 months) and mOS (23.5 vs. 18.0 months) vs. patients with *EGFR* 19del receiving chemo-immunotherapy [46,62].

VEGF was associated with NSCLC progression, recurrence, and metastasis [63]. Complimentarily with EGFR pathways, VEGF also promoted an immunosuppressive TIME [63], and anti-angiogenic therapy reprogrammed the TIME from immunosuppression to immune-supportive status [64,65]. Anti-angiogenic drugs plus chemotherapy were the most common regimen for EGFR-TKI-advanced NSCLCs [66]. Chemo-immunotherapy achieved a higher ORR than chemo-anti-angiogenesis therapy (29.5% vs. 13%) for EGFR-TKI-advanced NSCLCs [66]. Longer PFS was also associated with earlier anti-angiogenic drug applications in chemo-immunotherapy patients [61]. In a Phase III Impower150 study, chemo-immuno-anti-angiogenesis therapy was superior to either chemo-immunotherapy or chemo-anti-angiogenesis therapy on mPFS (10.2, 6.9, 6.9 months) and mOS (29.4, 19.0, 18.1 months) in a subgroup of EGFR-TKI-advanced *EGFR*-mutant NSCLCs [67]. However, around 66.7% of chemo-immuno-anti-angiogenesis patients experienced grade 3/4 treatment-related AEs [67]. In a Phase II clinical trial, 40 EGFR-TKI-advanced *EGFR*-mutant NSCLC patients were enrolled for chemo-immuno-anti-angiogenesis therapy, resulting in impressive mPFS (9.4 months), one-year OS (72.5%), and ORR (62.5%), with only 37.5% reporting irAEs [68].

## 3. Challenges

### 3.1. PD-L1 Has Limited Biomarker Roles in Immunotherapy in EGFR-Mutant NSCLCs

Expression of PD-1 on activated T cells, B cells, and natural killer (NK) cells blunted the immune response through interaction with its major ligand PD-L1, expressed on tumor cells and infiltrating immune cells [69,70,71]. Disruption of the PD-1/PD-L1 interaction reactivated the anti-tumor T cell-mediated cell cytotoxicity [69,70,71]. However, researchers did not consistently agree on PD-L1 expression levels in *EGFR*-mutant NSCLCs. In earlier studies, oncogenic EGFR signaling upregulated PD-L1 expression in *EGFR*-mutant NSCLC cell lines through the activated PI3K-AKT, MAPK-ERK, and/or JAK-STAT3 pathway [72,73,74,75,76]. EGFR TKI treatment decreased PD-L1 expression [72,73]. Some studies reported no correlation between PD-L1 expression and *EGFR* mutation status [77,78]. In more recent studies from clinical samples, the *EGFR*-mutant NSCLC group expressed significantly lower PD-L1 than the *EGFR* wild-type NSCLC group [36,79,80,81,82,83,84,85,86,87]. A pooled analysis of 15 public studies also suggested that *EGFR*-mutant NSCLCs had decreased PD-L1 expression [36]. PD-L1 expression was more accentuated at portions with higher PD-L1 expression in *EGFR*-mutant vs. *EGFR* wild-type group (51% vs. 68% at TC ≥ 1%, 8% vs. 35% at TC ≥ 25% and 5% vs. 28% at TC ≥ 50%) [42]. The EGFR blockade upregulated PD-L1 expression in *EGFR*-mutant NSCLCs [34,54,88].

PD-L1 was a predictive and prognostic biomarker for response to immunotherapy in NSCLCs [25]. Several clinical trials reported the association between PD-L1 expression and clinical outcomes in NSCLC patients [25,33,89]. However, the predictive and prognostic roles for PD-L1 in immunotherapy were limited in *EGFR*-mutant NSCLCs [39]. ICI therapy did not show clinical benefits in TKI-naïve *EGFR*-mutant NSCLCs, even with high PD-L1 expression. A Phase II trial of ICI therapy in treatment-naïve *EGFR*-mutant NSCLCs was discontinued due to lack of efficacy despite 73% of enrolled patients with more than 50% PD-L1 expression [33,89]. Patients with low or even undetectable PD-L1 expression also had improved survival with ICI therapy vs. chemotherapy [25]. Cross-comparison is sometimes challenging due to various immunohistochemical (IHC) assays with different scoring systems and cutoff values [90,91], making a universal assay for assessing PD-L1 expression with appropriate cutoff points important.

PD-L1 expression levels might not directly reflect the underlying T cell activity in *EGFR*-mutant NSCLCs [39]. PD-L1 was expressed in tumor and circulating immune cells, such as dendritic cells and myeloid-derived immune suppressor cells [92,93]. In POPLAR and OAK trials, assessing PD-L1 expression in tumor cells and tumor-infiltrating immune cells led to the finding that higher PD-L1 levels in both tumor cells and tumor-infiltrating immune cells were associated with improved patient survival after ICI therapy [25,94]. Patients with more than 30% PD-L1^+^CD11b^+^ myeloid cells before ICI therapy showed a 50% superior response rate [95]. Not only PD-L1 could bind with PD-1 to negatively regulate T cell activity, PD-L2 was another ligand for PD-1 with 2-6-fold higher affinity to PD-1 than PD-L1 [96,97]. EGFR signaling also regulated PD-L2 expression in NSCLCs [98].

### 3.2. TMB Is Low in EGFR-Mutant NSCLCs

TMB is the total number of gene alterations, including substitutions, insertions, and deletions. As TMB was associated with high levels of neoantigens [99], NSCLC patients with high TMB responded better to ICI therapy [100]. However, *EGFR* mutations were associated with decreased tumor mutation burden compared with tumors with wild-type *EGFR* [36,37,101]. Older people had increased TMB [99]. *EGFR* 19del was commonly found in the young, while *EGFR* L858R predominated in the elderly [5]. Consistently, lung tumors with *EGFR* 19del harbored a lower tumor mutation burden than *EGFR* L858R lung tumors [43,101]. In contrast to NSCLCs harboring common *EGFR* mutations, patients with uncommon *EGFR* mutations, especially the G719X mutation, showed the highest TMB (7.5 mutations/Mb), followed by *EGFR* exon 20 ins (4.6 mutations/Mb), *EGFR* T790M (4.05 mutations/Mb), *EGFR* L858R (3.4 mutations/Mb), and *EGFR* 19del (3.1 mutations/Mb) (*p* < 0.05) [37]. Like PD-L1, high TMB was detected in responders and non-responders receiving ICI therapy [102], indicating TMB was not the only determinant for ICI response.

### 3.3. EGFR-Mutant NSCLCs Are Non-Smokers, Generally

Tobacco is a known risk factor for lung cancer. However, PD-L1 positivity was also associated with smoking history [79,80,81], and smokers were more likely to benefit from ICI therapy [103,104,105]. Although *EGFR* mutations are more enriched in never-smokers, *EGFR* mutations in NSCLCs were also found in ever-smokers [6,106,107,108]. Compared with common *EGFR* mutations, uncommon *EGFR* mutations were significantly associated with smoking [109]. These findings further support the above-discussed findings that the NSCLCs with uncommon *EGFR* mutations were more likely to respond favorably to ICI therapy than NSCLCs with common *EGFR* mutations [49,53].

### 3.4. EGFR-Mutant NSCLCs Have a Lymphocyte-Depleted TIME

Crosstalk between cancer cells and TIME is a hot research topic with the rapid development of immunotherapy in cancers, including lung cancer. Based on the presence or absence of tumor-infiltrating lymphocytes (TILs), there were four different types of TIME in tumors: TIL^+^PD-L1^+^, TIL^-^PD-L1^−^, TIL^-^PD-L1^+^, and TIL^+^PD-L1^−^. However, only TIL^+^PD-L1^+^ tumors with lymphocyte infiltration and PD-L1 expression responded to ICI therapy [110]. To better reflect the complex relationship of the tumor, host, and environmental factors, tumors were also classified into the following types: the immune-desert tumor, the immune-excluded tumor, and the inflamed tumor [111]. The immune-desert and immune-excluded tumors were naturally resistant to ICI therapy. *EGFR*-mutant tumors had a “lymphocyte depletion” phenotype [34,36,38,112,113], characterized by a pronounced lack of the infiltration of CD8^+^ T cells [34,36,38,114], suggesting an immunosuppressive TIME. Single-cell RNA sequencing further confirmed the low T cell infiltration in the TIME of treatment-naïve *EGFR*-mutant NSCLCs [115]. Additionally, *EGFR*-mutant NSCLC tumors had markedly less crosstalk between T cells and other cell types via the PD-1/PD-L1 pathway than *EGFR*-negative NSCLCs [116].

Regulatory T cells (Tregs), especially the Forkhead box P3 (Foxp3)^+^ T cells, played essential roles in immune suppression [117]. *EGFR*-mutant NSCLC tumors showed a high infiltration of Foxp3^+^CD4^+^ regulatory T cells [38,118,119]. Retrospective immunohistochemistry analysis of 164 *EGFR*-mutant and 159 *EGFR* wild-type tumors revealed that the expression of CD3, CD4, and Foxp3 was significantly higher in *EGFR*-mutant NSCLC tumors than that in the *EGFR* wild-type NSCLC tumors [120]. The EGFR blockade increased intratumor CD8^+^ T cells and decreased Tregs infiltration in the TIME [38,121,122,123,124]. Like Tregs, myeloid-derived suppressor cells (MDSCs), known to suppress immune response [125,126,127], were also elevated in *EGFR*-mutant NSCLC tumors [124].

Tumors without detectable *EGFR* expression responded to EGFR inhibition [128], implicating that EGFR blockade potentially influenced the tumor-specific immune responses. Deficient *Egfr* in murine myeloid cells decreased carcinogenesis, suggesting a tumor-promoting function by myeloid cell-intrinsic EGFR signaling [129]. Macrophages in the TIME expressed *EGFR* [129,130]. EGF secreted by tumor cells promoted an M2 polarization of tumor-associated macrophages (TAMs), associated with the suppression of cytotoxic T cell function [131]. In contrast to M2-type TAMs, higher infiltrated M1-type TAMs found in NSCLC with uncommon *EGFR* mutations (G719X and exon 20s) correlated with longer PFS than common *EGFR* mutations [37].

*EGFR*-mutant NSCLC cell lines significantly downregulated MHC class I molecule expression compared with the *EGFR* wild-type NSCLC cell lines in response to IFNγ [132]. PI3K-AKT and MAPK pathways, the primary downstream signaling pathways of EGFR [133,134], also suppressed MHC class I molecule expression [132,135,136,137,138]. The data suggested that the downregulated MHC class I molecules were through abnormal EGFR signaling in *EGFR*-mutant NSCLCs. Indeed, EGFR inhibition augmented the expression of MHC class I molecules [139,140,141].

### 3.5. EGFR-Mutant NSCLCs Respond Poorly to ICI Therapy Alone or in Combination

*EGFR*-mutant NSCLCs generally responded poorly to ICI therapy [30,34]. In the Phase III CheckMate-057 clinical trial, subgroup analysis of the patients with activating *EGFR* mutations revealed no PFS and OS benefit from ICI therapy [23]. Subgroup analysis of Phase III KEYNOTE-010 clinical trial data showed no improved OS benefit from ICI therapy in *EGFR*-mutant NSCLCs [27]. Similarly, *EGFR*-mutant NSCLCs did not achieve prolonged OS from ICI therapy vs. chemotherapy in Phase III OAK clinical trial [25]. A retrospective analysis showed that *EGFR* mutations were associated with low clinical response to ICI blockade in NSCLCs [34]. It was confirmed in a pooled analysis from three clinical trials (CheckMate-057, KEYNOTE-010, and POPLAR) that the *EGFR* wild-type but not the *EGFR*-mutant NSCLCs had prolonged OS [31]. Combining data from five trials (CheckMate-017, CheckMate-057, KEYNOTE-010, OAK, and POPLAR), Lee et al. additionally confirmed no prolonged OS in *EGFR*-mutant NSCLCs receiving ICI therapy relative to chemotherapy [32]. A Phase II trial was halted due to lack of efficacy and two deaths within six months of enrollment to ICI therapy in treatment-naïve *EGFR*-mutant patients [33]. One of the two deaths was from pneumonitis [33]. In the Phase II WJOG8515L clinical trial, nivolumab was inferior to the chemotherapy with worse mPFS (1.7 vs. 5.6 months) and ORR (9.6% vs. 36%) in EGFR-TKI-advanced *EGFR*-mutant NSCLCs without a T790M mutation [142]. In a retrospective study with 58 *EGFR*-mutant NSCLCs, patients who responded to prior EGFR TKIs for more than ten months displayed significantly shorter PFS to ICI therapy compared with those who responded to prior EGFR TKIs for less than ten months (1.6 vs. 1.9 months, *p* = 0.009) [143]. Adding ICI therapy to chemotherapy was associated with worse survival than platinum doublet chemotherapy alone in osimertinib-advanced *EGFR*-mutant NSCLCs [144]. Phase III IMpower130 clinical trial also found no benefit in the *EGFR*-mutant subgroup treated with ICI therapy and chemotherapy combined vs. chemotherapy alone [145]. Based on all these negative findings, the National Comprehensive Cancer Network (NCCN) clinical practice guidelines of NSCLC (version 4, 2021) did not recommend immunotherapy for treating *EGFR*-mutant NSCLCs.

### 3.6. Safety Concerns and Lower Clinical Outcomes Regarding EGFR TKI and ICI Combined for the Treatment of EGFR-Mutant NSCLCs

Compared with chemotherapy, ICI therapy generally correlates with fewer adverse reactions. However, ICI therapy was associated with immune-related adverse events (irAEs), probably due to the disruption of immunologic homeostasis [146]. There is a growing concern that a combination of EGFR TKIs, especially osimertinib or gefitinib, and ICI therapy may be associated with an increased risk of toxicity. In the Phase 1/2 KEYNOTE-021 clinical trial, five of seven untreated stage IIIB/IV *EGFR*-mutant NSCLC patients (71.4%) treated with ICI therapy plus gefitinib had grade 3/4 liver toxicity, leading to permanent treatment discontinuation in four patients [147]. A lung adenocarcinoma patient bearing *EGFR* 19del treated with osimertinib following ICI therapy had Stevens–Johnson syndrome and hepatotoxicity [148]. In a Phase Ib TATTON clinical trial, 38.5–60% of EGFR-TKI-advanced *EGFR*-mutant NSCLCs experienced grade 3/4 irAEs, and 30–40% were discontinued to osimertinib and ICI therapy combined due to irAEs [149]. Concurrent osimertinib and ICI therapy were associated with an increased incidence of irAEs, leading to the early termination of a Phase III CAURAL recruitment [150]. Interstitial lung disease (ILD) was a severe adverse response to EGFR TKIs [151]. Osimertinib treatment that immediately followed prior ICI therapy also caused a high incidence of ILD [152,153]. A meta-analysis of eight studies further concluded that a higher chance of irAEs was observed in EGFR-TKI plus ICI therapy in EGFR-TKI-advanced *EGFR*-mutant NSCLCs [154].

Other than toxicity concerns, worse clinical outcomes were seen in *EGFR*-mutant NSCLCs treated with EGFR TKI and ICI combined than those treated with EGFR TKI alone. The Phase Ib TATTON clinical trial showed that osimertinib and ICI combination therapy only achieved 43% ORR in the EGFR-TKI-advanced *EGFR*-mutant NSCLCs [149]. In the Phase III CAURAL clinical trial, decreased ORR was also seen in the osimertinib plus ICI group vs. osimertinib alone [150]. Similarly, the ORR in gefitinib plus ICI therapy (14.3%) was much worse than the 55% RR in gefitinib alone for *EGFR*-mutant NSCLC patients [155]. In contrast to gefitinib plus ICI therapy, the untreated stage IIIB/IV *EGFR*-mutant NSCLCs tolerated erlotinib plus ICI therapy and better ORR (41.7%) in the KEYNOTE-021 [147]. The 41.7% ORR was much lower than erlotinib alone (~70%) [156,157].

### 3.7. Hyper-Progressive Disease (HPD)

Hyper-progressive disease (HPD), characterized by unexpected fast tumor enlargement, both at rate and volume and early fatality of patients [158,159], was observed in 13.8% of NSCLC patients during treatment with ICI therapy in a retrospective study [160]. By contrast, around 20% of *EGFR*-mutant NSCLCs showed risk of HPD after ICI therapy [159].

## 4. Conclusions

So far, ICI-based immunotherapy is ineffective for treatment-naïve *EGFR*-mutant NSCLC patients. Whether EGFR-TKI-advanced *EGFR*-mutant NSCLC patients may benefit from the ICI-based immunotherapy is worth further investigation, considering there is no approved effective treatment for this population. IrAEs will be an unmet roadblock during the evaluation. IrAEs may indicate the immune system is awakened. However, it is hard to predict irAEs, and some strong irAEs, if not controlled, may be deadly. A better understanding of the relationship of irAEs and efficacy may be helpful for future clinical monitoring during immunotherapy application. Other than irAEs, many other questions are still unsolved. Who may benefit from the immunotherapy: PD-L1 TPS > 50%, smoking history, high tumor mutation burden, high CD8^+^ T cells, and/or specific subtypes of *EGFR* mutations? Immunotherapy is not limited to anti-PD-1/PD-L1 monoclonal antibodies and could go beyond them. Additionally, the combination of ICI-based immunotherapy and several other treatment modalities is under active investigation.

## Data Availability

Not applicable.

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
