# Peer review of "Hope and Challenges: Immunotherapy in EGFR-Mutant NSCLC Patients"

_biomedicines, 2023, doi:10.3390/biomedicines11112916_

Round 1
Reviewer 1 Report
Comments and Suggestions for Authors
I read the article titled 'It is too early to say NO IMMUNOTHERAPY for EGFR-mutant NSCLC patients' with interest.
It is well well-researched and easy-to-read article. This is quite a good collection of information about the ineffectiveness of ICIs in EGFR (+) patients, taking into account the role of PD-L1, irAEs and the microenvironment immunological status of the tumor. The author collected interesting. I have no major objections to it, except that the title is too strong concerning the entire article and conclusion because in general, the article mainly concerns ICI failures in EGFR(+) patients described in the literature and clinical trial data.
Author Response
Appreciate your kind and positive evaluation. Thank you for your suggestion on the title. The ICI therapy generally failed in treatment-naïve EGFR-mutant patients is mainly discussed in the challenge part. I also think that the treatment-advanced EGFR-mutant patients and some NSCLC patients with uncommon EGFR mutations may potentially get better help from ICI therapy than chemotherapy as discussed in the Hope part. As suggested, the title was updated as “Hope and challenges: immunotherapy in EGFR-mutant NSCLC patients”.
Reviewer 2 Report
Comments and Suggestions for Authors
This is a well-prepared Review on the role of immunotherapy in patients with EGFR positive NSCLC.
The author has prepared a thorough review of the literature on the topic and presents available immunotherapy treatment options. The EGFR mutations treatments with TKI are also presented. Existing studies on survival and disease free survival are explained.
Author Response
Thank you for your positive evaluation.